# Stop Wasting My Time!
# Saving Days of ImageNet and BERT Training with Latest Weight Averaging

**Jean Kaddour**
Centre for Artificial Intelligence
University College London
`jean.kaddour.20@ucl.ac.uk`

## Abstract

Training vision or language models on large datasets can take days, if not weeks. We show that averaging the weights of the $k$ latest checkpoints, each collected at the end of an epoch, can speed up the training progression in terms of loss and accuracy by dozens of epochs, corresponding to time savings up to ~68 and ~30 GPU hours when training a ResNet50 on ImageNet and RoBERTa-Base model on WikiText-103, respectively. We also provide the code and model checkpoint trajectory to reproduce the results and facilitate research on reusing historical weights for faster convergence[1].

## 1 Introduction

The arsenal of deep learning methods (e.g., architectures, regularizers, pre-trainers, etc.) has been growing rapidly; for the last decade, thousands of them have been proposed yearly. Arguably, many are brittle and not as universally effective as initially claimed [24]. One way to filter *"what really works"* is by testing methods on large datasets. For example, for vision tasks, methods that demonstrated success on ImageNet have often proven to be successful in other tasks [1].

Large datasets, however, require access to expensive multi-GPU machines to enable data parallelism and reasonable training durations. Less well-funded researchers do not have access to supercomputers, and lengthy training runs make quick, iterative experimentation of research ideas difficult. Simple task-, model-, and optimizer-agnostic methods that can be easily added to existing training pipelines and speed up training time have the potential to make deep learning research more accessible.

In the 90s, Polyak & Juditsky [33] studied how to accelerate the convergence speed of stochastic gradient descent in the convex loss function regime. They proved that the running average of the model weights iterates $\bar{\boldsymbol{\theta}} = \frac{1}{t}\sum_{i=1}^{t}\boldsymbol{\theta}_i$ converges to the loss minimizer $\boldsymbol{\theta}^*$ asymptotically with the highest possible rate. When visualizing a convex loss function, the geometric intuition is simple: whenever the optimizer oscillates around a minimum, the average of the iterates will be closer to it.

However, in deep learning, loss functions are highly non-convex [8]. Weight averaging has been mainly used to improve the model's generalization performance at the end of or after training [14, 16].

**Contribution** We revisit weight averaging applied to neural networks from a convergence speed perspective. Inspired by Li et al. [22], we focus on the *middle* stage of training: after the dramatic changes of the local loss landscape during the very first training steps [10, 7] but *before* the optimizer converges. Because the weights still undergo substantial change in that middle phase, averaging *all*

---

[1] https://github.com/jeankaddour/lawa

Has it Trained Yet? Workshop at the Conference on Neural Information Processing Systems (NeurIPS 2022).

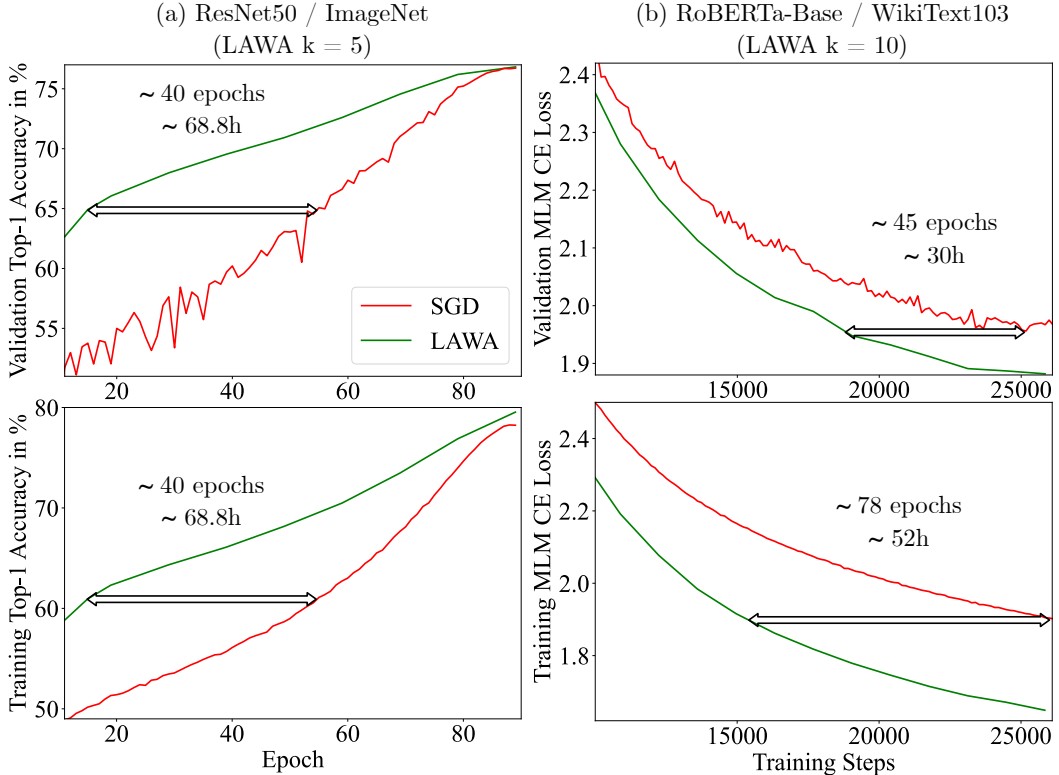

(a) ResNet50 / ImageNet
(LAWA k = 5)

(b) RoBERTa-Base / WikiText103
(LAWA k = 10)

Figure 1: **LAWA speeds up convergence on ImageNet and (Ro)BERT(a) training**. We highlight the GPU hours saved by LAWA as the longest time until the baseline optimizer (SGD or Adam) matches LAWA 's performance. We plot the losses and top-5 accuracies in Appendix A.

collected models, e.g., by maintaining a moving average [14, 16], can be sub-optimal. Therefore, we propose to average the $k$ latest checkpoints (each collected at the end of an epoch) throughout training, which we refer to as *LAtest Weight Averaging* (LAWA).

## 2 LAtest Weight Averaging (LAWA)

**The key idea** is to collect model checkpoints once at the end of each epoch in a queue. LAWA 's solution at the end of epoch $E$ is $\boldsymbol{\theta}_E^{\text{LAWA}} := \frac{1}{k} \sum_{i=E-k+1}^{E} \boldsymbol{\theta}_i$.

**Requirements** include few training loop modifications, as shown in Algorithm 1, and additional memory. In practice, we store the checkpoints in RAM or on disk and only transfer them to the GPU once we want to evaluate $\boldsymbol{\theta}^{\text{LAWA}}$. To improve the time complexity of the averaging operation, one can use a circular queue [2].

**The number of latest weights** $k$ is a hyper-parameter, and we achieve good results across both experiments with default value $k = 6$, as shown in Figure 2. However, we observe that averaging too many checkpoints ($k > 16$) results in worse performance.

**Algorithm 1** Pseudocode in PyTorch style

```
# k: number of latest checkpoints
ckpts = []
lawa_model = copy.deepcopy(model)
for epoch in range(num_epochs):
    for (x,y) in train_loader:
        train_step(x, y, model, optimizer)
    ckpts.append(get_params(model))
    if epoch + 1 >= k:
        update_lawa_model(lawa_model,
            ckpts)
        del ckpts[0]

def update_lawa_model(lawa_model, ckpts):
    for p, avg_p in zip(list(lawa_model.
        parameters()),torch.mean(ckpts)):
        p.copy_(avg_p)
```

**The averaging coefficients** can also follow a different pattern, e.g., in Appendix C, we experiment with an exponential moving average (assigning higher weights to the more recent checkpoints) and

---

[2]Coding interview preparers might remember this Leetcode problem.

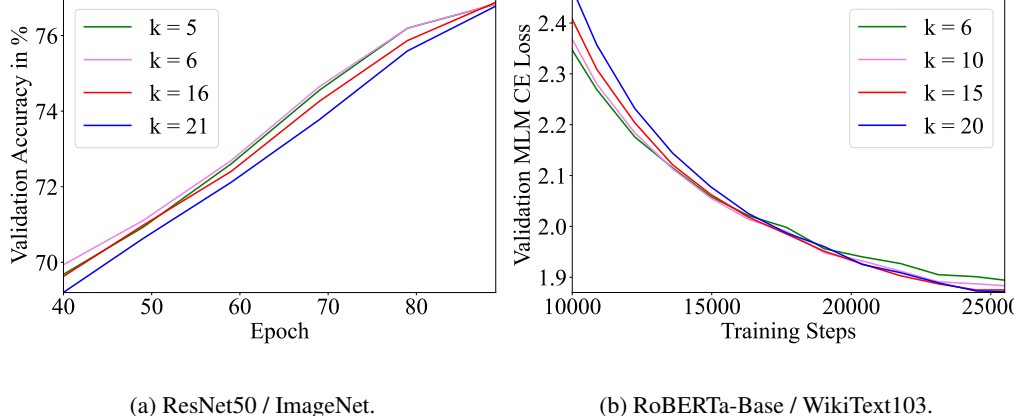

(a) ResNet50 / ImageNet.
(b) RoBERTa-Base / WikiText103.

Figure 2: **LAWA is fairly insensitive w.r.t.** $k$, but high $k$ can hurt its performance early on.

observe that this works worse than uniform coefficients. One can also learn the averaging coefficients [38, 22], but for simplicity and to avoid additional computational costs, we do not do so.

**The checkpoint saving frequency** might be thought of as a hyper-parameter; however, in this work, we always set it to one epoch. When there is so much data that we are in a sub-one-epoch training regime [18, 13], we may collect checkpoints every $\nu$ steps and need to tune $\nu$. Another heuristic might be to collect a checkpoint whenever the validation loss has not improved [28].

**If the network includes batch norm layers,** then their statistics for $\theta_E^{\text{LAWA}}$ are unknown. Prior work [14] has suggested computing them by an inference pass through the training dataset. We do not observe a large effect of doing so compared to simply copying $\theta_E$'s statistics, possibly because we only average the $k$ latest weights instead of keeping one running average over many epochs [14].

## 3 Results

We run all experiments on a machine with 4x NVIDIA 3090s and report its wall-clock time.

### 3.1 Image Classification: ResNet50 on ImageNet

We consider the ImageNet 1000-classes classification task [4], which includes 1.28M training images and 50k validation images. To train a ResNet50 [12], we use the official PyTorch implementation [32] and train for 90 epochs using SGD with a momentum value of 0.9 and a cosine learning rate schedule. Our 4-GPU machine takes ~26min for one epoch. For $\theta^{\text{LAWA}}$, we re-compute the batch norm layer statistics with a full inference pass through the training dataset before evaluating $\theta^{\text{LAWA}}$.

In Figure 1(a), we observe that LAWA reaches a high accuracy dramatically faster, e.g., validation accuracy of around ~66% (the final accuracy is ~76%) is reached ~40 epochs (~68 hours) earlier than the baseline optimizer (SGD). However, we also note that its head start decreases towards the end of the training, and the highest reached accuracy is not reached much earlier. This observation raises the question of whether we can use LAWA to "jump forward" and continue the training from $\theta^{\text{LAWA}}$ to reach the optimal accuracy faster, which we further discuss in Section 4.

### 3.2 Masked Language Modeling: RoBERTa-Base on WikiText-103

Next, we pre-train a (Ro)BERT(a)-Base [5, 25] model with masked language modeling (MLM) objective on the WikiText-103 dataset [27] with 103M and 218k tokens for training and validation set, respectively. We follow the training recipe provided by `fairseq` [31]: We train with Adam [20] for 200 epochs, using a batch size of 2048, a polynomial learning rate decay with 10k warmup steps and a peak learning rate of 0.0005. Our 4-GPU machine needs ~10min for one training epoch.

In Figure 1(b), we report the training and validation MLM cross-entropy (CE) losses as a function of the number of training steps (as typically done in NLP). We observe that LAWA consistently

improves the losses, and it reaches Adam's final best validation loss ~45 epochs ahead, saving ~30 GPU hours. Interestingly, $\theta^{\text{LAWA}}$'s final validation performance is noticeably better than $\theta^{\text{ADAM}}$'s, confirming previous results on improved generalization obtained with weight averaging [14, 16].

## 4 Future Work

**Continuing training from $\theta^{\text{LAWA}}$.** It is tempting to think that we may "jump forward" training by applying the LAWA procedure and then continue training from there if some target accuracy has not been reached yet. One issue is that we would need to adjust the learning rate each time we "jump". In practice, we may not know by how much (if at all) we accelerated the training progression. Hence, it remains unclear how to adjust a learning rate scheduler or the state variables of an adaptive optimizer.

$k$ **scheduler.** In Figure 2, we observe that at different times, different $k$ values perform better; e.g., during the end of the training, higher $k$ performs better; motivating a scheduler for $k$.

**Accelerating training from the very beginning.** We focus on speeding up training during the middle stage of training: after the first training steps but long before the optimizer converges. The reason for that is that in the very early training phase, the gradient typically moves with large magnitudes until it converges to a smaller subspace of the loss function's Hessian, in which it then remains over long periods of training [10, 7] (middle stage). We empirically confirm that averaging during the early phase worsens the baseline's performance, as can be seen in Figure 4.

**Combining LAWA with other acceleration techniques.** As we will discuss in the next section, there are several other techniques available to accelerate neural network training. For example, the SAM optimizer [6] can accelerate training too [37], and Kaddour et al. [16] show that SAM combined with weight averaging can further boost the final test performance.

**Relationships between LAWA and optimization hyper-parameters.** For example, SGD becomes unstable for certain learning rates [39, 15]; can we similarly characterize when LAWA is effective?

**Applying other operations to a set of checkpoints.** For example, by learning a hyper-network [11] that takes in one or more checkpoints and predicts the model parameters at later training stages.

**When does it not work?** LAWA may not always cause speed-ups because Kaddour et al. [16] reported some negative results on using weight averaging to improve the model's final performance.

## 5 Related work

The idea of weight averaging is not novel; it has been studied widely in linear settings [33, 30, 21].

Szegedy et al. [35] used weight averaging to create the GoogLeNet model, which, at that time, set a new state of the art in the ImageNet 2014 challenge [4]. Izmailov et al. [14] introduce *Stochastic Weight Averaging* (SWA), a weight averaging strategy starting from pre-trained models to move them to better-generalizing regions in the same loss basin. Kaddour et al. [16] extensively study SWA's effectiveness, including non-typical domains like graph-structured data, and suggest combining it with SAM [6] to boost its final performance further. Wortsman et al. [38] propose to average weights of multiple models with different hyper-parameter configurations. All three works (i) average weights toward the end or even after convergence, (ii) focus on the models' final test performances, and (iii) incorporate one moving-averaged model, while we show in Figure 2 that too large $k$ can result in suboptimal results, especially at earlier training times.

This work is heavily inspired by Li et al. [22]'s *Trainable Weight Averaging* (TWA), who propose to learn averaging coefficients for training speed-ups. Concurrently, Guo et al. [9] observe that running the SWA procedure multiple times accelerates convergence. In some sense, LAWA generalizes their procedure by keeping an average of the $k$ latest checkpoints instead of running SWA sequentially. Another related optimizer utilizing an auxiliary set of "fast weights" before updating the weights of interest is the *Lookahead* (LA) optimizer [41]. We compare LAWA and LA in Appendix B.

Another line of work has shown that training data re-weighting can speed up training. For example, some re-weighting methods focus on proxy models [2, 29], importance sampling [3, 19] or removing spurious correlations [17, 36].

## Acknowledgements

I thank Matt J. Kusner and Mingtian Zhang for feedback and fruitful discussions. I acknowledge support from the Engineering and Physical Sciences Research Council with grant number EP/S021566/1.

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

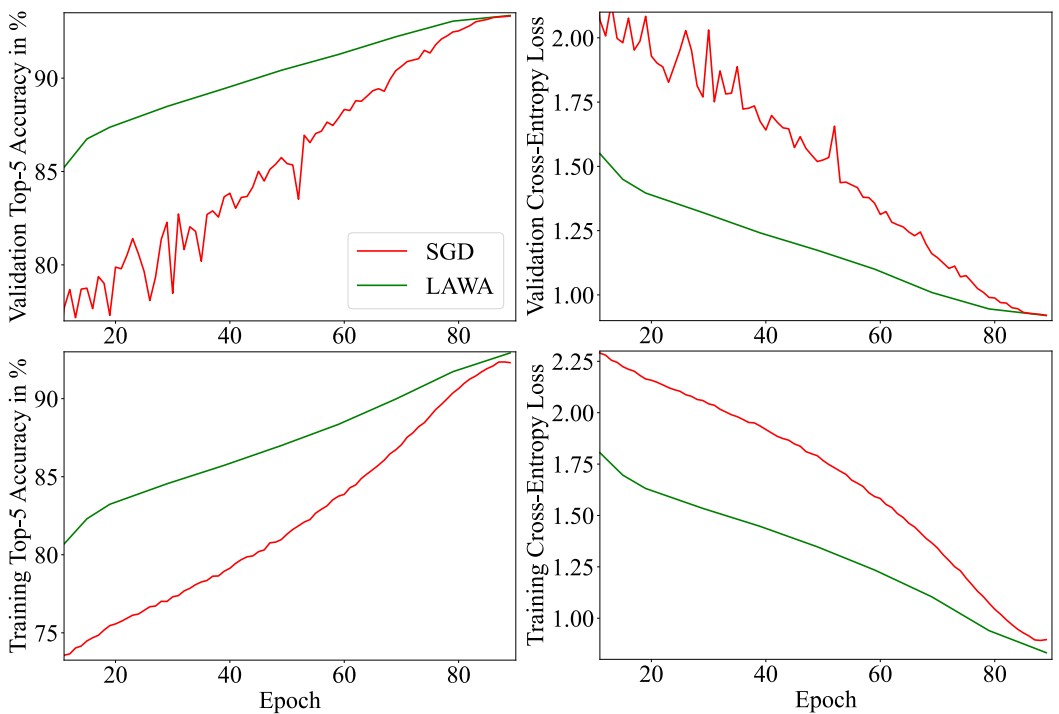

Figure 3: **Additional ResNet50 / ImageNet Metrics:** training/validation top-5 accuracies and losses.

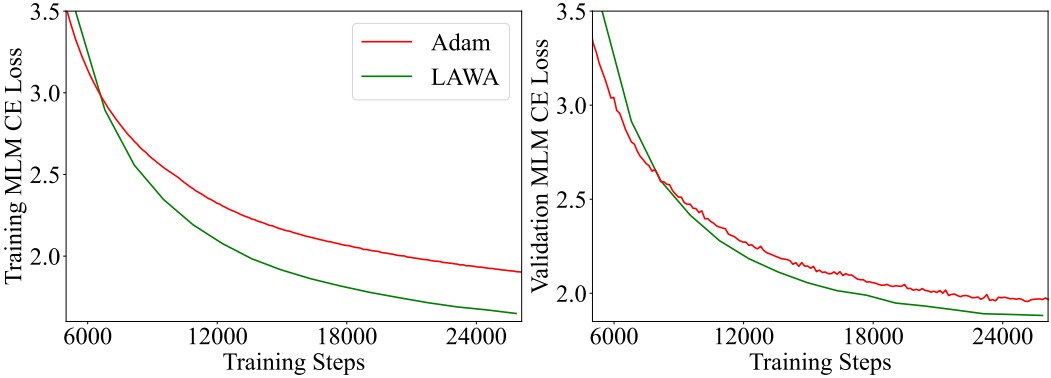

Figure 4: **Starting point of LAWA being effective:** on RoBERTa-Base / WikiText103.

# A   Losses and Top-5 Accuracies

For completeness, we also plot the training and validation losses for both experiments and the top-5 accuracies for the ImageNet experiment.

Figure 3 shows similar speed up trends of LAWA over SGD as discussed in the main body (Figure 1).

Figure 4 shows the training and validation losses for RoBERTa-Base trained on WikiText103. Here, we also include losses during earlier stages of training and point out that during these more fluctuant phases, LAWA performs worse. We expect this behavior because previous works pointed out that the network undergoes dramatic changes in early phases [34, 10, 7].

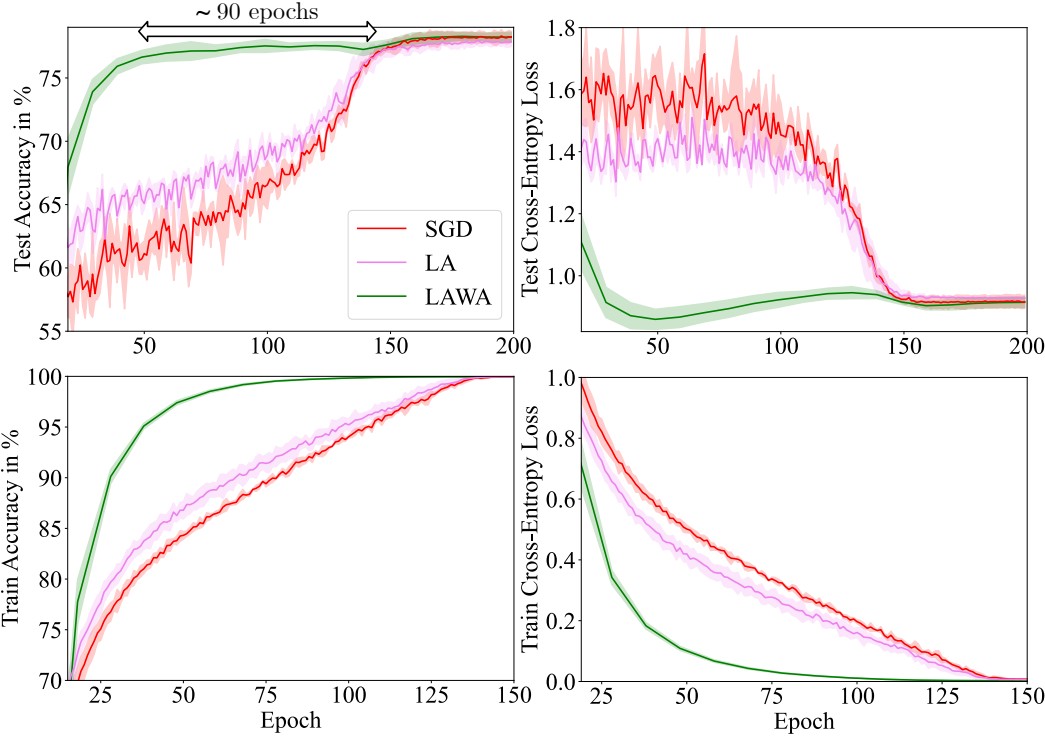

Figure 5: **LAWA ($k = 10$) outperforms Lookahead [41]**. We plot the mean and standard deviation of ResNet34/CIFAR-100 experiments across three random seeds.

## B  LAWA vs. Lookahead

We compare LAWA ($k = 10$) against Lookahead [41] on the moderately-sized CIFAR-100 dataset (50k training images) and train a ResNet34 [12]. We follow commonly used hyper-parameters (see e.g., [40, 23]), and train with SGD for 200 epochs, using a batch size of 256, a momentum value of 0.9 and a cosine learning rate scheduler [26, 6] with initial learning rate $\eta = 0.1$. For LA, we use $\alpha = 0.8$ and $k_{LA} = 5$, as suggested by the authors for this particular CIFAR100 dataset.

Figure 5 shows the training/test accuracy/loss as a function of the number of epochs. LAWA reaches high test accuracy around 90 epochs earlier than SGD/LA.

Initially, we started experimenting with this learning task before scaling up to larger datasets. Since we only observed slight but not dramatic improvements in LA over the baseline, we did not evaluate LA in the larger-scale ImageNet and BERT experiments. However, note that we apply LAWA to the SGD checkpoints; an interesting future direction can be to combine LAWA with LA, i.e., to average over checkpoints obtained with LA.

## C  Uniform vs. Exponentially Decayed Averaging Coefficients

We compare uniform (UNI, corresponding to $\boldsymbol{\theta}^{\text{LAWA}}$ by default) and exponentially-decaying (EXP) weight coefficients. We follow the same ResNet34 / CIFAR100 setup as in the previous section.

For EXP, we set $\alpha = 0.9$ and compute

$$\boldsymbol{\theta}_0^{\text{EXP}} = \boldsymbol{\theta}_0, \quad \forall E > 0 : \boldsymbol{\theta}_E^{\text{EXP}} := \alpha \boldsymbol{\theta}_E + (1 - \alpha) \boldsymbol{\theta}_{E-1}^{\text{EXP}}. \tag{1}$$

We set $k = 10$ for both strategies. Figure 6 shows that UNI slightly outperforms EXP; however, the difference is not large.

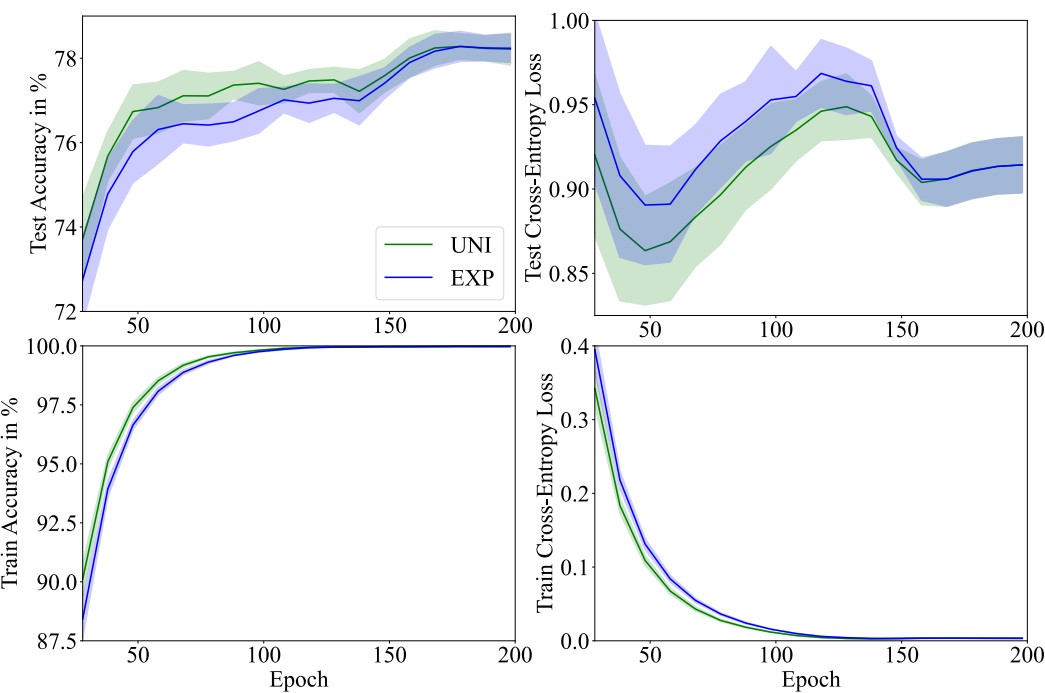

Figure 6: **Uniform coefficients (i.e., LAWA by default) slightly outperform exponentially-decaying ones for** $k = 10$. We plot the mean and standard deviation of ResNet34/CIFAR-100 experiments across three random seeds.

