# OpenReview forum: "Stop Wasting My Time! Saving Days of ImageNet and BERT Training with Latest Weight Averaging"
_NeurIPS.cc/2022/Workshop/HITY — HITY Workshop NeurIPS 2022_

### Official Review · Reviewer_w4aY · 2022-10-06
**Applying weight averaging to the last $k$ checkpoints during training provides better any-time performance.**

**Rating:** 1
**Confidence:** 3

**Review:**

The paper proposes an algorithm called LAWA (LAtest Weight Averaging) which keeps a separate model that is an average of the parameters of the latest $k$ model checkpoints. This LAWA model provides a significantly better predictive performance in the middle of the training process.

The experiments are presented clearly and show that LAWA provides better predictions in the middle of training compared to the baselines.

Feedback:
- Perhaps it would be better to not talk about training speed improvements or that LAWA "speeds up convergence" (caption of Figure 1). LAWA doesn't reduce the time it takes these models to converge. However, LAWA improves the any-time performance of a model (while still in training). This shouldn't diminish the contributions of the papers, but I think would be more precise.

---

### Official Review · Reviewer_H5h6 · 2022-10-07
**Middle-phase training improvements via weight averaging**

**Rating:** 1
**Confidence:** 3

**Review:**

This paper demonstrates that averaging the weights of a vision or language model over the last $k$ checkpoints can speed up training in the middle phase. The introduced method is called LAtest Weight Averaging (LAWA) and the paper explores different degrees of freedom, e.g. number of averaged checkpoints and type of average operation. LAWA's benefits are measured in time savings of GPU hours. The paper also discusses directions for future work.

---

I think the paper would benefit from a short "Conclusion" section at the end that restates the most important findings.

---

Miscellaneous comments:
- Fig. 2 is referenced before Fig. 1. Maybe swap them?

---

### Decision · Program_Chairs · 2022-10-20

Accept